# Rapid learning and unlearning of predicted sensory delays in self-generated touch

**Konstantina Kilteni\*, Christian Houborg, H Henrik Ehrsson**

Department of Neuroscience, Karolinska Institutet, Stockholm, Sweden

**Abstract** Self-generated touch feels less intense and less ticklish than identical externally generated touch. This somatosensory attenuation occurs because the brain predicts the tactile consequences of our self-generated movements. To produce attenuation, the tactile predictions need to be time-locked to the movement, but how the brain maintains this temporal tuning remains unknown. Using a bimanual self-touch paradigm, we demonstrate that people can rapidly unlearn to attenuate touch immediately after their movement and learn to attenuate delayed touch instead, after repeated exposure to a systematic delay between the movement and the resulting touch. The magnitudes of the unlearning and learning effects are correlated and dependent on the number of trials that participants have been exposed to. We further show that delayed touches feel less ticklish and non-delayed touches more ticklish after exposure to the systematic delay. These findings demonstrate that the attenuation of self-generated touch is adaptive.

DOI: https://doi.org/10.7554/eLife.42888.001

## Introduction

It is theorized that the brain uses internal models to anticipate the sensory consequences of voluntary movements on the basis of a copy of the motor command (efference copy) (**Wolpert and Flanagan, 2001**; **Bays and Wolpert, 2007**; **Franklin and Wolpert, 2011**; **Blakemore et al., 1998a**). These sensory predictions are used to achieve efficient online motor control, to ensure movement stability and to reduce uncertainty, as the actual sensory feedback is delayed due to sensory transduction times (**Kawato, 1999**; **Davidson and Wolpert, 2005**; **Shadmehr and Krakauer, 2008**). In addition, the predictions of the internal models are also used for attenuating the perception of self-produced input, thereby increasing the salience of unpredicted external signals and facilitating the perceptual distinction between the self and the environment (**Blakemore et al., 2000a**; **Bays and Wolpert, 2008**). For example, when one actively touches one's left hand with one's right (self-touch), the touch feels less intense than identical touches applied to the left hand by another person or a machine (**Blakemore et al., 2000a**; **Shergill et al., 2003**; **Blakemore et al., 1999**). This is because the self-induced touch has been predicted by the internal model.

Importantly, the predictions of the internal models are useful only if the models constitute accurate representations of the body and the current environmental dynamics (**Wolpert and Flanagan, 2001**; **Shadmehr and Krakauer, 2008**; **Shadmehr et al., 2010**; **Wolpert et al., 1998**; **Miall, 2002**; **Haith and Krakauer, 2013**). Biased predictions would not only be detrimental to motor performance but also prevent one from distinguishing the sensory feedback of one's own movements from that produced by external causes. A good illustration of this sharp tuning of the internal models is that their predictions are temporally locked to the given movement (**Figure 1a**, left): for example, during self-touch, tactile feedback is expected at the time of contact between the hands, and touch that is artificially delayed (even by only 100 ms) shows reduced attenuation and is attributed to external causes rather than the self (**Bays et al., 2005**; **Blakemore et al., 1999**).

**\*For correspondence:**
konstantina.kilteni@ki.se

**Competing interests:** The authors declare that no competing interests exist.

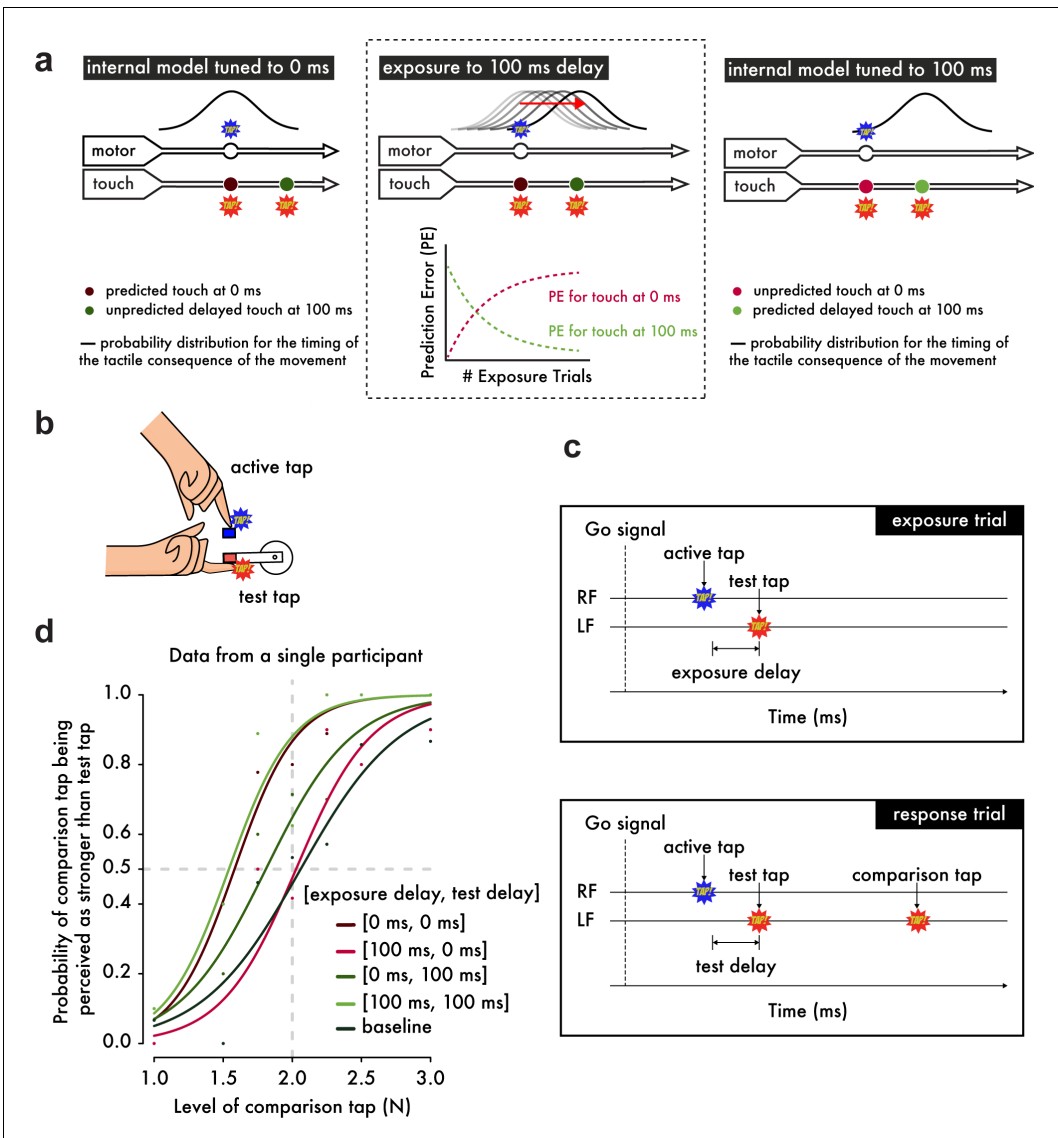

**Figure 1.** Experimental Hypotheses, Procedures and Psychophysical Analysis. (**a**) (*Left*) When the internal model is tuned to 0 ms as in natural situations, the probability distribution for the occurrence of touch on the left index finger (approximated as a normal distribution (*Bays et al., 2005*) based on the uncertainty in the predicted time of the touch arising from noise in motor and sensory systems) peaks at 0 ms after the movement of the right index finger. Touch presented at 0 ms shows the strongest attenuation, while touch at 100 ms is less attenuated because it is less likely to have been self-generated. (*Middle*) When exposed to systematic delays of 100 ms between the finger movement and the touch, the model parameter is gradually updated to 100 ms, which can be viewed as a simple incremental shift in the probability distribution by 100 ms. Before the exposure, there is an error associated with the touch predicted at 0 ms and presented at 100 ms but no error related to the naturally presented touch at 0 ms. During the learning period, this pattern gradually reverses: a prediction error for the touch presented at 0 ms appears and grows over exposure time, while the prediction error for the touch at 100 ms decays and reaches a minimum. (*Right*) After prolonged exposure, the touch at 0 ms has low probability, produces a large prediction error and will not be attenuated, whereas the touch at 100 ms has high probability, produces no prediction error and will be attenuated. (**b**) Participants were instructed to use their right index finger to tap a sensor (active tap) that delivered a tap on their left index finger (test tap). (**c**) In the *exposure trials*, participants simply tapped the sensor with their right index finger (RF) and received the tap on the left index finger (LF) with a 0 ms or a 100 ms *exposure delay* (intrinsic delay of the force setup ≅ 36 ms). In the *response trials*, participants received a second tap on their left index finger (comparison tap) and were required to indicate which tap was stronger: the test or the comparison tap. The test tap could be presented with a *test delay* of either 0 ms or 100 ms. (**d**) Psychophysical data from a representative participant demonstrate how the somatosensory attenuation phenomenon is

*Figure 1 continued on next page*

*Figure 1 continued*

quantified. The horizontal gray dashed line indicates the 50% point of psychometric functions, and the vertical gray dashed line indicates the true intensity of the test tap (2 N).

DOI: https://doi.org/10.7554/eLife.42888.002

Here, we demonstrate that the brain can rapidly (a) unlearn to expect touch at the moment of contact between the hands and (b) learn to predict delayed touch instead. Using a device that simulates bimanual self-touch (*Figure 1b*), thirty subjects were initially exposed to 500 trials in which a systematic delay of 0 ms or 100 ms (*exposure delay*) was inserted between the voluntary tap of the right index finger and the resulting touch on the pulp of the relaxed left index finger (*Stetson et al., 2006*) (*Figure 1c*). We reasoned that when repeatedly presented with the 100 ms discrepancy between the predicted and actual somatosensory feedback, the brain would be forced to retune the internal model in order to account for this delay and thus keep the predictions accurate (*Figure 1a*, middle). This hypothesis led to two specific predictions (*Figure 1a*, right). First, when the 100 ms delay is removed after the exposure period, participants should have stopped predicting and therefore attenuating the sensation of the tap. Second, when the delay is maintained after the exposure period, the participants should have started predicting and thus attenuating the delayed tap. We tested both of these predictions in a psychophysical task (*Bays et al., 2005*) performed immediately after the initial exposure (*Figure 1d*) (see also Materials and methods).

## Results and discussion

In the response trials of the task, participants were presented with two taps on the left index finger – one test tap of 2 N presented at 0 ms or 100 ms after the right finger's active tap (*test delay*) and one comparison tap of variable magnitude – and their task was to indicate which one felt stronger (*Figure 1c*). The points of subjective equality (PSEs) extracted from the psychophysical curves represent the attenuation of the test tap and are displayed in *Figure 2a* for each pair of exposure and test delay. A no-movement condition where the participants simply relaxed their right hand served as a baseline for basic somatosensory perception.

As expected from previous studies (*Bays et al., 2005*; *Blakemore et al., 1999*), when participants were exposed to a 0 ms delay, attenuation was observed only for the immediate test tap (paired t-test between [exposure delay = 0 ms, test delay = 0 ms] and baseline, $t(29) = -4.65$, p<0.001, $CI^{95} = [-0.195, -0.076]$, $BF_{10} = 369$) and not for the delayed touch (paired Wilcoxon test between [0 ms, 100 ms] and baseline, $n = 30$, $V = 226$, p=0.903, $CI^{95} = [-0.061, 0.056]$). In line with this pattern, the immediate tap felt significantly less intense than the delayed one (paired Wilcoxon test between [0 ms, 0 ms] and [0 ms, 100 ms], $n = 30$, $V = 18$, p<0.001, $CI^{95} = [-0.183, -0.09]$). Importantly however, the pattern of results reversed after exposure to a 100 ms delay: the attenuation of the immediate tap significantly decreased (paired t-test between [100 ms, 0 ms] and [0 ms, 0 ms], $t(29) = 2.73$, p=0.011, $CI^{95} = [0.018, 0.128]$, $BF_{10} = 4.241$). This decrease in attenuation of immediate touch (*Figure 2c*) indicates that participants *unlearned* to predict touch at the time of contact. Moreover, the attenuation of the immediate tap was no longer significantly different from the baseline (paired t-test between [100 ms, 0 ms] and baseline, $t(29) = -1.88$, p=0.070, $CI^{95} = [-0.129, 0.005]$) although the Bayesian analysis did not provide compelling evidence regarding the presence or absence of such difference ($BF_{10} = 0.916$). In contrast, the attenuation of the delayed tap significantly increased (paired t-test between [100 ms, 100 ms] and [0 ms, 100 ms], $t(29) = -3.29$, p=0.003, $CI^{95} = [-0.142, -0.033]$, $BF_{10} = 14.311$). This shift in attenuation of the delayed touch (*Figure 2d*) indicates that participants *learned* to predict the touch at the delay to which they were exposed. Moreover, a statistical trend for a difference in the attenuation of the delayed tap from the baseline was detected (paired t-test between [100 ms, 100 ms] and baseline, $t(29) = -1.98$, p=0.057, $CI^{95} = [-0.153, 0.002]$) but without any conclusive evidence from the Bayesian analysis ($BF_{10} = 1.077$). Importantly, the extent to which participants unlearned to predict the immediate touch was significantly positively correlated with the extent to which participants learned to predict the delayed one (Pearson's $r = 0.471$, $t(28) = 2.83$, p=0.009, $CI^{95} = [0.136, 0.712]$, $BF_{10} = 6.150$; *Figure 2b*), implying a temporal shift in the probability distribution of the tactile consequences in line with our hypothesized model (*Figure 1a*, middle and right). Finally, we noted that there were no significant differences in the

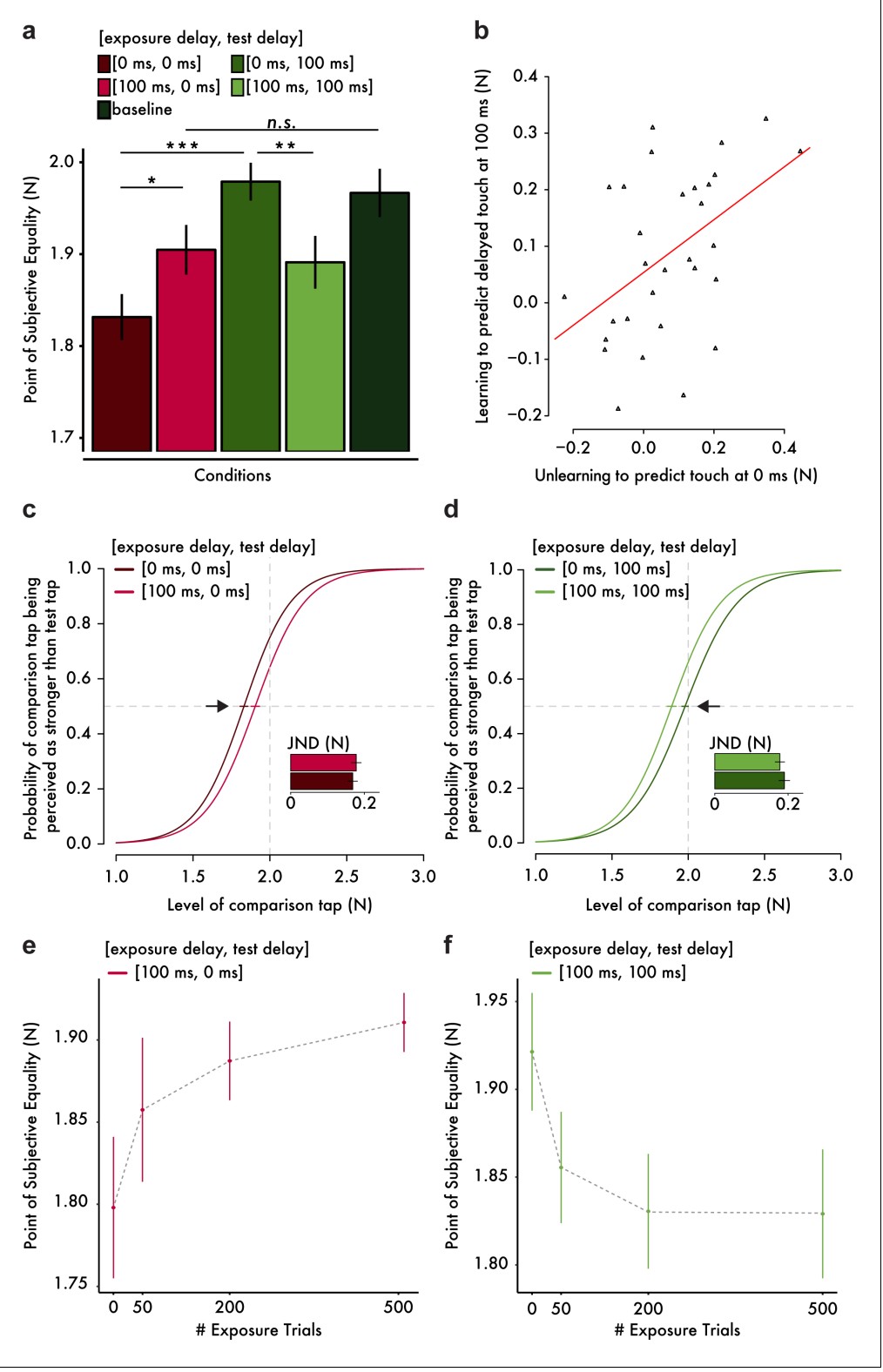

**Figure 2.** Results from the somatosensory attenuation experiments. (**a**) Mean PSE (± s.e.m.) for each condition. Only the important planned comparisons are displayed (*p<0.05, **p<0.01, ***p<0.001, *n.s.* not significant). (**b**) Scatterplot of the attenuation shifts in immediate touch (unlearning) and delayed touch (learning). The more participants unlearned to predict the immediate touch, the more they learned to predict the delayed one. The orange line indicates the fitted regression line. (**c, d**) Group psychometric functions indicating significant

*Figure 2 continued on next page*

*Figure 2 continued*

attenuation shifts for immediate (**c**) and delayed touch (**d**). Error bars indicate the standard error of the mean (s.e. m.) of the PSEs. Subplots indicate the mean JND (± s.e.m.) for each condition. The horizontal gray dashed lines indicate the 50% point of psychometric functions, and the vertical gray dashed lines indicate the true intensity of the test tap (2 N). (**e, f**) Mean PSE (± s.e.m.) as a function of the number of exposure trials.

DOI: https://doi.org/10.7554/eLife.42888.003

The following source data and figure supplement are available for figure 2:

**Source data 1.** Mean PSE (± s.e.m.) for each condition.
DOI: https://doi.org/10.7554/eLife.42888.005
**Source data 2.** Attenuation shifts in immediate touch (unlearning) and delayed touch (learning).
DOI: https://doi.org/10.7554/eLife.42888.006
**Source data 3.** Model parameters for the group fits.
DOI: https://doi.org/10.7554/eLife.42888.007
**Source data 4.** Mean PSE (± s.e.m.) as a function of exposure trials.
DOI: https://doi.org/10.7554/eLife.42888.008
**Figure supplement 1.** Individual Fits of all participants in Experiment 1.
DOI: https://doi.org/10.7554/eLife.42888.004

participants' discrimination ability, that is just noticeable difference, between conditions (paired t-test between [0 ms, 0 ms] and [100 ms, 0 ms], $t(29) = -0.76$, p=0.452, $CI^{95} = [-0.035, 0.016]$, $BF_{10} = 0.254$); paired t-test between [0 ms, 100 ms] and [100 ms, 100 ms], $t(29) = 0.82$, p=0.418, $CI^{95} = [-0.018, 0.043]$, $BF_{10} = 0.265$)). This finding excludes the presence of response sensitivity differences between conditions as an alternative explanation of the present results. Taken together, these findings suggest that the internal model generating the tactile predictions that produce the somatosensory attenuation can be temporally retuned.

To quantitatively strengthen our conclusion that the abovementioned findings are due to the retuning of the internal model, we performed an additional experiment in which we explicitly tested our theoretical prediction that the longer the participants are exposed to the systematic delays between movement and touch, the larger the temporal shift in the probability distribution of the internal model will be, and therefore the larger the perceptual effects on somatosensory attenuation will be (*Figure 1a*, middle). Two new groups of fifteen participants each were continuously presented with 100 ms exposure trials while being tested for the attenuation of immediate (*Figure 2e*) or delayed touch (*Figure 2f*) after 0, 50, 200 and 500 exposure trials. The results revealed a significant interaction between the number of exposure trials and the delay tested ($F(3,56) = 3.89$, p=0.013). More specifically, the attenuation of immediate touch [100 ms, 0 ms] decreased significantly as the number of exposure trials increased ($F(3,42) = 3.64$, p=0.020), while the attenuation of delayed touch [100 ms, 100 ms] increased significantly as the number of exposure trials increased ($F(3,42) = 4.86$, p=0.005). Replicating our previous results, the pairwise comparisons indicated a significant decrease in the attenuation of immediate touch after 200 exposure trials ($t(14) = -2.45$, p=0.028, $CI^{95} = [-0.168, -0.011]$, $BF_{10} = 2.403$) and 500 exposure trials ($t(14) = -2.82$, p=0.014, $CI^{95} = [-0.198, -0.027]$, $BF_{10} = 4.344$) compared to the initial performance after 0 exposure trials, as well as a significant increase in the attenuation of the delayed touch after 50 exposure trials ($t(14) = 2.98$, p=0.010, $CI^{95} = [0.019, 0.113]$, $BF_{10} = 5.627$), 200 exposure trials ($t(14) = 2.82$, p=0.014, $CI^{95} = [0.022, 0.160]$, $BF_{10} = 4.324$) and 500 exposure trials ($t(14) = 3.38$, p=0.004, $CI^{95} = [0.034, 0.151]$, $BF_{10} = 10.833$) compared to the initial test. Collectively, these results demonstrate that the retuning of the internal model underlying somatosensory attenuation occurs through error-driven processes that evolve over time in response to repeated exposure to unexpected delays in the sensorimotor system.

Somatosensory attenuation is considered one of the reasons for which we cannot tickle ourselves (*Blakemore et al., 1998b*). Accordingly, self-tickling sensations are cancelled because the somatosensory feedback of our movement matches the tactile prediction of the internal model and thus gets attenuated. In contrast, ticklish sensations arise from discrepancies (prediction errors) between the predicted feedback of the internal model and the actual somatosensory input (*Blakemore et al., 2000a*). An earlier study showed that participants rated their self-generated touch as more ticklish when a delay greater than 100 ms was introduced between the movement of one hand and the

resulting touch on the other, compared to when a 0 ms delay was introduced. We hypothesized that after exposure to systematic delays the delayed self-generated touch would feel less ticklish because the retuning of the internal model of somatosensory attenuation would reduce this delay-induced prediction error (*Figure 1a*, middle). Conversely, natural (non-delayed) self-generated touch would feel more ticklish since the prediction error between the delayed prediction and the immediate tactile feedback would increase after the exposure.

To this end, we performed an additional experiment in which a new group of thirty participants moved the arm of a robot with their right hand to apply touch on their left forearm through a second robot. The second robot (slave) copied the movement of the first robot (master) either with a 0 ms or a 150 ms delay (intrinsic delay of the robots' setup $\cong$ 44 ms) (*Figure 3a*). As expected from the literature, after exposure to the 0 ms delay participants judged more frequently the delayed touch as being more ticklish than the immediate touch (median frequency = 0.8, mean frequency = 0.73). Critically, after exposure to the 150 ms delay, this frequency significantly dropped (median frequency 0.6, mean frequency 0.65): Wilcoxon singed rank test, $n = 30$, $V = 88$, p=0.046, $CI^{95} = [-0.2,-0.00002]$ (*Figure 3b*). That is, the delayed touch was rated significantly less frequently as the more ticklish one, or, reversely, the immediate touch was rated significantly more frequently as the more ticklish one. This result suggests that ticklishness sensations depend on the same learning mechanism that supports the attenuation of self-touch, thereby generalizing our findings beyond force intensity perception and suggesting a universal role of the sensory predictions –generated by a continuously retuned internal model– in the perceptual discrimination of self and non-self.

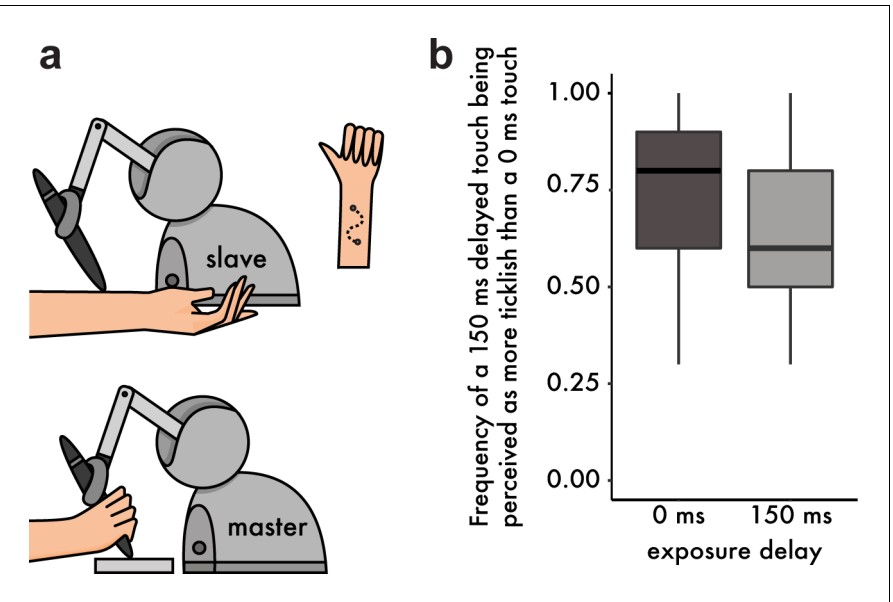

**Figure 3.** Methods and Results from the tickling experiment. (**a**) Participants were instructed to move the stylus of a robot with their right hand (master robot). The stylus was free to move along a sinusoidal path within a 3D-printed mold, thus constraining the movement trajectory. A second robot (slave) copied the master robot and delivered the touch on the volar (anterior) part of their left forearm. Participants were asked to perform two sinusoidal movements (twice back and forth) with their right hand and received two stimulations on the left forearm (one with a 0 ms and one with a 150 ms delay). Afterwards they had to indicate which touch felt more ticklish. (**b**) Boxplot of the frequencies at which participants rated the delayed touch (150 ms) as more ticklish than the immediate touch. After exposure to the 150 ms delay, participants chose the delayed touch less frequently (or reversely, they chose the immediate touch with higher frequency). The horizontal black bars represent the medians, and the boxes denote the interquartile ranges.

DOI: https://doi.org/10.7554/eLife.42888.009

The following source data is available for figure 3:

**Source data 1.** Median and interquartile range (IQR) for the frequency of a 150 ms delayed touch being perceived as more ticklish than a 0 ms touch, per condition.
DOI: https://doi.org/10.7554/eLife.42888.010

The present study investigated the temporal retuning of the internal model underlying the perceptual attenuation of self-generated touch, the latter being a well-established index of the efference-copy-based sensory predictions (*Wolpert and Flanagan, 2001*; *Bays et al., 2005*; *Blakemore et al., 2000a*; *Bays and Wolpert, 2008*; *Kilteni et al., 2019*). Our findings are strongly consistent with a gradual updating of the internal model during exposure to systematic delays between the movement and the tactile feedback from the resulting self-touch (*Figure 1a*). After exposure to such delays, the delayed touch was predicted and thus attenuated, while the immediate (non-delayed) touch was not predicted and thus not attenuated. Moreover, the magnitudes of these effects were correlated and dependent on the number of exposure trials, in line with our proposal that the retuning of the internal model was driven by a prediction-error-based learning process. Finally, we demonstrated that this dynamic retuning of the internal model influences the perceived ticklishness of tactile stimulation, so that delayed touches feel less ticklish and non-delayed touches more ticklish after exposure to systematic delays. This demonstrates that the predictive learning process under discussion affects the perceptual quality of touch as being self- or externally generated beyond the mere intensity of the somatosensory feedback. Taken together, the present study brings compelling evidence that somatosensory attenuation is an adaptive phenomenon.

We propose that the internal model underlying somatosensory attenuation during normal temporal conditions is dynamically retuned in presence of systematic delays, to encode the new temporal relationship between the motor command and the tactile consequence. Rather than the adaptation of an existing internal model, it could be argued that the exposure to the delays leads to the acquisition of a new internal model instead. In a bimanual object manipulation task, *Witney et al. (1999)* demonstrated a significant grip force modulation after repeated exposure to a systematic delay between the movement of one hand and the resulting effects on the other hand, in a direction that is consistent with the acquisition of new internal model rather than the update of an existing one. According to this proposal, the brain would learn different internal models for the different delays and would switch between them (*Wolpert and Kawato, 1998*). Our data cannot differentiate between these two hypotheses since in both scenarios we would expect a decrease in the attenuation of the immediate touch and an increase in the attenuation of delayed touch. Moreover, it is not known whether the same internal model underlies the anticipatory grip force modulation during object manipulation and the sensory attenuation during self-touch. Nevertheless, based on the correlation between the shifts in the attenuation of immediate touch and the attenuation of delayed touch (*Figure 2e–f*), we consider that a shift in the predicted temporal distribution towards the newly predicted timing (update of an existing internal model) is more likely than the acquisition of a new one.

Our study goes beyond earlier studies on crossmodal lag adaptation and sensorimotor temporal recalibration (see *Chen and Vroomen, 2013*; *Rohde and Ernst, 2016* for reviews). By employing the self-touch paradigm, we kept the relationship between the movement and its feedback from the body natural, in contrast to previous studies that provided participants with artificial feedback, for example visual flashes on the screen (*Stetson et al., 2006*; *Cai et al., 2012*) or white noise bursts (*Heron et al., 2009*) in response to keypresses. Most importantly, rather than measuring changes in the perceived order of the participants' movements and those associated external events – effects present in mere crossmodal asynchronies (*Vroomen et al., 2004*; *Fujisaki et al., 2004*; *Hanson et al., 2008*) in the absence of movement– we measured the attenuation of self-generated touch that requires active movement (*Bays et al., 2005*; *Kilteni et al., 2019*) and thus, efference-copy-based predictions.

What could be the neural mechanism of the temporal retuning of the internal model underlying somatosensory attenuation? We speculate that a candidate brain area could be the cerebellum, given its involvement in the implementation of the internal models (*Wolpert et al., 1998*), its well-established relationship with motor learning (*Wolpert et al., 2011*) and somatosensory attenuation (*Blakemore et al., 1998b*; *Kilteni and Ehrsson, 2019*) and its relevance to time perception and temporal coordination of movement according to evidence from nonhuman primates (*Ashmore and Sommer, 2013*), cerebellar patients (*Ivry and Keele, 1989*) and healthy subjects (*Jueptner et al., 1995*). Given that maintaining unbiased sensorimotor predictions is the root of motor learning (*Shadmehr et al., 2010*; *Wolpert et al., 2011*; *Krakauer and Mazzoni, 2011*), we propose that the observed effects are a new form of learning that represents the updating of the internal model's parameter specifying the temporal relationship between the motor command and its sensory

feedback from the body. In contrast to the classically studied motor (force-field and visuomotor) adaptation paradigms (*Shadmehr et al., 2010*; *Krakauer and Mazzoni, 2011*), the present learning occurs in the temporal rather than the spatial domain, and it serves to keep the sensorimotor predictions regarding one's own body (*Kilteni and Ehrsson, 2017*) temporally tuned. This space-time analogy becomes more apparent when we consider that our observed temporal aftereffects – that is the reduced attenuation observed during 0 ms delay after exposure to the 100 ms – mirror the classic spatial aftereffects (e.g. reaching errors) observed in force-field and visuomotor adaptation after removing the spatial perturbation participants have been exposed to (*Shadmehr et al., 2010*).

The adaptive nature of sensory attenuation has broad implications for cognitive neuroscience beyond motor control. Accurate comparisons of predicted and actual sensory feedback underpin the distinction of self and environment (*Frith et al., 2000*; *Blakemore and Frith, 2003*; *Crapse and Sommer, 2008*; *Fletcher and Frith, 2009*), the perception of ticklishness (*Blakemore et al., 1999*) and the sense of agency, that is the sense of being the author of voluntary action (*Haggard, 2017*). The present results reveal how such fundamental cognitive distinctions between the self and external causes are supported by a dynamic and flexible error-driven learning process. Indeed, the data from our tickling experiment submit that it is possible to learn to tickle oneself: after the retuning of the internal model to the delay, natural (non-delayed) self-touch feels more intense and more ticklish, as an external touch does.

Our findings on learning and unlearning mechanisms could have important clinical relevance for schizophrenia research. In healthy subjects, reduced sensory attenuation was associated with a high tendency towards delusional ideation (*Teufel et al., 2010*), and the frequency of passivity experiences in non-pathological subjects with high schizotypal traits was related to increased ticklishness ratings for self-produced touch (*Lemaitre et al., 2016*). Similarly, schizophrenic patients (*Shergill et al., 2005*; *Shergill et al., 2014*) and patients with auditory hallucinations and/or passivity experiences (*Blakemore et al., 2000b*) were shown to exhibit reduced attenuation of their self-generated touches with respect to matched controls, with the severity of the schizophrenic patients' hallucinations being a predictor of their reduced somatosensory attenuation (*Shergill et al., 2014*). This relationship between somatosensory attenuation and schizophrenia was proposed to reflect deficits in the patients' internal models mechanisms (*Frith, 2005*). Specifically, *Whitford et al. (2012)* proposed that schizophrenia is related to an abnormal myelination of frontal white matter that produces delays in the generation of the predicted consequences based on the efference copy (internal model). That is, the predicted timing of the sensory feedback lags the movement and feedback time and therefore, non-delayed feedback (0 ms) comes before its predicted time and it is not attenuated, thereby producing uncertainty about the origin of the signal (the self or the others). In agreement with this view, a study using encephalography showed that schizophrenic patients exhibited reduced cortical suppression of self-generated sounds when these were presented without delay but normal attenuation when presented with a delay, compared to heathy controls (*Whitford et al., 2011*). Accordingly, we theorize that schizophrenic patients would perceive their delayed touch as less intense and less ticklish, reflecting an internal model erroneously tuned at that delay; a prediction that should be tested in future experiments.

More fundamentally, since our study suggests that sensory attenuation relies on online updating prediction estimations it opens up for the possibility that it could be this learning process that is impaired in schizophrenia. We therefore speculate that schizophrenic patients might have no problem in generating motor commands or generating sensory predictions as such, but it is the continuous updating of these predictions that is impaired. In other words, rather than a structural change per se causing the changes in sensory attenuation, it might be that it is the inability to retune the internal model to compensate for these changes in the brain that is causing the cognitive deficits and psychiatric symptoms. We therefore propose that future computational psychiatry research should specifically investigate the capacity of these individuals to learn and unlearn new temporal relationships between their movements and their sensory feedback.

# Materials and methods

## Materials

In all conditions, participants rested their left hand, palm up, with the left index finger placed inside a molded support. Participants received forces on the pulp of their left index finger from a cylindrical probe (20 mm diameter) that was attached to a lever controlled by a DC electric motor (Maxon EC Motor EC 90 flat; manufactured in Switzerland). The right hand and forearm were comfortably placed on top of boxes made of sponge, with the right index finger resting on top of a force sensor (FSG15N1A, Honeywell Inc.; diameter, 5 mm; minimum resolution, 0.01 N; response time, 1 ms; measurement range, 0–15 N). The force sensor was placed on top of (but not in contact with) the cylindrical probe that was contacting their relaxed left index.

A screen blocked the participants' view of their hands and forearms during all conditions, and the participants were further asked to fixate their gaze on a fixation cross marked at two meters across from them. In addition, participants were wearing headphones through which white noise was administered so that no sound created either by the motor or by the right hand's tap could be used as a cue for the psychophysics task. An auditory cue (tone) served to indicate to participants when to press the force sensor with their right index finger during the task.

## Participants

After providing written informed consent, thirty naïve participants (15 women and 15 men, 27 right-handed and three ambidextrous [*Oldfield, 1971*]) aged 18–32 years participated in the experiment. The sample size was decided based on previous studies (*Stetson et al., 2006*; *Cai et al., 2012*).

## Conditions and procedures

The experiment included five conditions: four movement conditions and one baseline (no-movement) condition. Each movement condition included both *exposure* and *response* trials. The baseline condition assessed the participants' somatosensory perception of two successive taps on the left index finger without any movement of the right index finger and thus included only response trials.

### Exposure trials

On each exposure trial (*Figure 1c*), participants tapped the force sensor with their right index finger (active tap) after an auditory cue. This tap triggered the test tap on their left index finger. The test tap could be presented either with a 0 ms delay – therefore simulating self-touch – or with a 100 ms delay (*exposure delay*).

### Response trials

In each response trial (*Figure 1c*), as in the exposure trials, participants tapped the force sensor after the auditory cue and received the test tap on their left index finger with a delay of either 0 ms or 100 ms (*test delay*). After a random delay of 800–1500 ms from the test tap, a second tap (comparison tap) was delivered to the left index finger, and participants were required to indicate using a foot pedal which tap (the test tap or the comparison tap) was stronger. The test tap was always set to 2 N, while the intensity of the comparison tap was systematically varied among seven different force levels (1, 1.5, 1.75, 2, 2.25, 2.5 or 3 N). Trials at which the test tap did not have the proper intensity (2 N) – because for example the participants moved their left index finger during the tap – were rejected from the analysis. Specifically, we allowed a range between 1.85 N and 2.15 N. According to this criterion, we rejected 296 trials out of 11550 total trials (2.6%). The intensity of the comparison taps was always assigned to the closest value from the predefined intensities (for example, a comparison tap of 2.61 N was assigned the 2.5 N value). Both test and comparison taps had a fixed duration of 100 ms each. In a short session just before the experiment, we taught participants how to tap the sensor with their right index finger to prevent them from pressing too forcefully or too gently during the experiment. This psychophysical task has been previously validated to assess the magnitude of somatosensory attenuation (*Bays et al., 2005*).

The four movement conditions corresponded to the four combinations of exposure and test delay levels: [0 ms, 0 ms], [0 ms, 100 ms], [100 ms, 0 ms], and [100 ms, 100 ms]. Each movement condition consisted of 500 initial exposure trials (to 0 ms or 100 ms) and 70 response trials (each of the 7

intensities of the comparison tap was repeated 10 times). Each response trial followed five re-exposure trials, resulting in 850 exposure trials in total, per condition. The baseline condition consisted of 105 response trials (each of the 7 intensities of the comparison tap was repeated 15 times).

No feedback was ever provided to subjects with respect to their performance on the psychophysical task. The order in which the volunteers participated in the conditions was randomized.

### Data and statistical analysis

For each condition, we used a logistic regression model to fit the proportion of the participants' responses that the comparison tap was stronger than the test tap (*Equation 1*, *Figure 1d*):

$$f(x) = \frac{1}{1 + \exp(-(a + \beta x))} \tag{1}$$

where $\alpha$ represents the intercept and $\beta$ represents the slope. We used the function *glm* with a *logit* link function in the software R version 3.4.4 (*R Development Core Team, 2019*). We extracted the point of subjective equality ($PSE = -\frac{a}{\beta}$), which corresponds to the intensity of the comparison tap at which the participant perceives the test tap (2 N) and the comparison tap as equal ($p = 0.5$). Furthermore, we extracted the just noticeable difference ($JND = \frac{log3}{\beta}$), an index of the participant's response sensitivity.

We checked the normality of the distributions of PSEs and JNDs with Shapiro-Wilk tests. Accordingly, we performed planned comparisons with either a paired t-test or a Wilcoxon signed-rank test. To test for perceptual shifts due to the 100 ms exposure delays, we contrasted the conditions featuring a 100 ms exposure delay with those featuring a 0 ms exposure delay for the same test delay. We also compared all the movement conditions with the baseline condition. To calculate the correlation coefficient between the attenuation shift of immediate touch (unlearning, i.e., PSE at [100 ms, 0 ms] - PSE at [0 ms, 0 ms]) and the attenuation shift of delayed touch (learning, i.e., PSE at [0 ms, 100 ms] - PSE at [100 ms, 100 ms]), we used the Pearson correlation coefficient, since the data were normally distributed. All statistical tests were two-tailed. As mentioned above, for the group psychometric functions (*Figure 2c–d*), we generated the plots using the mean PSE and the mean JND across the thirty participants. Finally, for all parametric tests, we used JASP (*JASP Team, 2019*) to perform Bayesian analysis using default Cauchy priors with a scale of 0.707 in order to provide information about the level of support for the experimental hypothesis compared to the null hypothesis ($BF_{10}$) given the data.

### Participants, procedure and analysis in the effect of exposure experiment

We performed an additional experiment in which a new set of 30 naïve volunteers (12 women and 18 men, all 30 right-handed [*Oldfield, 1971*]) aged 18–35 years participated after providing written informed consent. Each participant was exposed to 500 trials with a systematic delay of 100 ms. Half of the participants were tested for the attenuation of the immediate touch [100 ms, 0 ms], and the other half were tested for the attenuation of the delayed touch [100 ms, 100 ms]. The psychophysical tests were performed at the beginning of the experiment (no initial exposure) and then at three time points spaced at intervals of 50, 150, and 300 exposure trials. That is, participants performed the psychophysical task after 0, 50, 200 and 500 cumulative initial exposure trials. One response (out of seventy) was missing for one participant in one psychophysical test. There were 304 rejected trials corresponding to 3.6% of the total number of trials (8400). We performed a mixed analysis of variance (ANOVA) on the PSEs with time (number of exposure trials) as the within-participants factor and the test delay as the between-participants factor. Then for each experiment separately, we performed a repeated-measures analysis of variance (ANOVA) on the PSEs with time as the within-participants factor. We then performed planned comparisons using paired t-tests, since all distributions satisfied the assumption of normality.

### Participants, procedure and analysis in the tickling experiment

A new set of 30 naïve volunteers (16 women and 14 men, 28 right-handed, 1 ambidextrous and one left-handed [*Oldfield, 1971*]) aged 20–38 years participated in the Tickling experiment after providing written informed consent. Two robots (Touch Haptic Devices, 3D systems, https://www.

3dsystems.com/haptics-devices/touch) were placed in front of the participants with a distance of 10 cm between them. Participants rested their left arm palm up within a forearm support made of sponge, just beneath the stylus of a robot (*slave* robot). The tip of the stylus was covered with sponge to reduce its sharpness. Participants rested their right elbow on an arm support and held with their right hand the stylus of another robot (*master* robot). The lower part of the stylus of the master robot could freely move within a sinusoidal 3D printed path. The distal point of the path (the one closest to the hand) served as a starting point. Each trial lasted three seconds and upon an auditory cue, participants were asked to move the stylus from the distal point (*Figure 3a*) to the proximal point along the path, and back to the distal point. In two different conditions, participants performed 50 exposure trials with a systematic delay of 0 or 150 ms between the movement of the master robot and the movement (and stimulation) of the slave robot (*exposure delay*). In the response trials, as in the exposure trials, participants performed two consecutive trials (the first with a 0 ms and the second with a 150 ms delay). Immediately afterwards, they were asked to report which of the two stimulations on their left forearm (the first or the second) felt more ticklish to them. As in the previous experiments, each response trial followed five re-exposure trials. There were 10 response trials per condition. The order in which the volunteers participated in the conditions was counterbalanced. We calculated the frequency at which participants judged the second (150 ms) touch as more ticklish in the two conditions. Since the data were bounded, we performed the planned comparison using a Wilcoxon signed-rank test.

In the tickling experiment, we chose a delay of 150 ms and not a delay of 100 ms as in the sensory attenuation experiments. This was because our pilot tickling experiments indicated that a 100 ms delay was not sufficient to differentiate the perception of a delayed stroke on the arm from that of an immediate stroke. We consider that this asymmetry of the delay sensitivity between attenuation and tickling should not be surprising: a 100 ms delay would be more salient for a self-induced tap of 100 ms duration than a continuous stroke of 3 s duration.

## Intrinsic delays in the experimental setup

As a technical side note, we measured the intrinsic delays of both experimental setups. For the force-setup, we recorded the delay between a tap applied by the right index finger on the force sensor and the corresponding tap the motor produced in response on the left index finger, in 20 consecutive trials. The estimated delay was: (mean ± SD) 36.2 ± 9.8 ms. Therefore, the experimental conditions labeled '0 ms' and '100 ms' actually correspond to effective delays of 36 ms and 136 ms. Similarly, for the robots' setup we recorded the difference between the time at which the stylus of the master robot reached the furthest position along a straight path along the depth axis and the time at which the stylus of the slave robot was at the corresponding position, in 20 consecutive paths. The estimated delay was: (mean ± SD) 44.2 ± 19.1 ms. Therefore, the experimental conditions labeled '0 ms' and '150 ms' in the tickling experiment actually correspond to effective delays of 44 ms and 194 ms. In both cases, the presence of the intrinsic delays does not alter the interpretation of the results since it only implies a further shift of the initial distribution at a time point later than the movement onset (*Figure 1a*, left).

## Acknowledgements

Konstantina Kilteni was supported by the Marie Skłodowska-Curie Intra-European Individual Fellowship (#704438). The project was funded by the Swedish Research Council, the Göran Gustafssons Stiftelse and the Torsten Söderbergs Stiftelse.

## Additional information

### Funding

| Funder | Grant reference number | Author |
| --- | --- | --- |
| H2020 Marie Skłodowska-Curie Actions | Marie Sklodowska-Curie Individual Fellowship #704438 | Konstantina Kilteni |
| Swedish Research Council | | H Henrik Ehrsson |

| Torsten Söderberg Foundation | Torsten Söderbergs Stiftelse | Christian Houborg |
| --- | --- | --- |
| Göranssonska Stiftelserna | | H Henrik Ehrsson |

The funders had no role in study design, data collection and interpretation, or the decision to submit the work for publication.

## Author contributions
Konstantina Kilteni, Conceptualization, Data curation, Software, Formal analysis, Supervision, Funding acquisition, Investigation, Visualization, Methodology, Writing—original draft, Writing—review and editing; Christian Houborg, Investigation, Methodology, Writing—review and editing; H Henrik Ehrsson, Conceptualization, Resources, Data curation, Supervision, Funding acquisition, Methodology, Writing—original draft, Project administration, Writing—review and editing

## Author ORCIDs
Konstantina Kilteni (iD) https://orcid.org/0000-0002-6887-6434
H Henrik Ehrsson (iD) https://orcid.org/0000-0003-2333-345X

## Ethics
Human subjects: Written informed consent was obtained from all participants. The Regional Ethical Review Board of Stockholm approved all three studies (ref: 2016/445-31/2 and 2018/2519-32).

## Decision letter and Author response
Decision letter https://doi.org/10.7554/eLife.42888.016
Author response https://doi.org/10.7554/eLife.42888.017

## Additional files
### Supplementary files
• Transparent reporting form DOI: https://doi.org/10.7554/eLife.42888.011

### Data availability
All datasets come from human subjects. We do not have ethical permit to make the raw individual data publicly available. The data used to generate Figures 2 and 3 is provided.

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
