## [Decision Letter]

**Acceptance summary:**

This study provides an interesting extension to the work on sensory attenuation, the experience that the sensed force of an actively produced tap is felt as less intense compared to when the tactile stimulus is not the result of a volitional action. Previous studies have demonstrated that this effect is predictive, based on cancellation of the expected sensory input – and one would therefore predict it should be adaptive to changes in the contingency between action and sensation, but this has not previously been demonstrated. The experimental design to explore this question is appropriate and the results convincingly show that this attenuation process is subject to temporal adaptation. This study extends our understanding of the adaptive nature of the nervous system on a rapid timescale, with our sensory experience modified to optimize information based on the current context.

**Decision letter after peer review:**

Thank you for submitting your article "Rapid learning and unlearning of predicted sensory delays in self-generated touch" for consideration by *eLife*. Your article has been reviewed by two peer reviewers, with Richard Ivry serving as the Reviewing and Senior editor. The following individual involved in review of your submission has agreed to reveal his identity: Marc Ernst (Reviewer #1).

The reviewers have discussed the reviews with one another and I have drafted this decision to help you prepare a revised submission.

Summary:

We all found the study to provide an interesting extension to the work on sensory gating, the experience that the sensed force of an actively produced tap is usually felt less intense than when the tactile stimulus is not the result of a volitional action. Previous studies have demonstrated that this effect is predictive, based on cancellation of the expected sensory input – and one would therefore predict it should be adaptive to changes in the contingency between action and sensation, but this has not previously been demonstrated. The experimental design to explore this question is appropriate and the results convincingly show that this gating process is subject to temporal adaptation. The finding that, after adaptation, the tactile stimulus is felt as less intense when presented at the adapted delay compared to when presented simultaneously, is especially compelling. The generalization work was also viewed as a strength of the paper, increasing the impact value.

There are a few issues that should be addressed in revision. We don't foresee any of the problems being difficult to address. They mostly relate to the statistical analyses. These could, in principle, affect the conclusions that can be drawn. However, we anticipate that the data are robust enough to withstand more conservative statistical tests. This needs to be demonstrated.

Essential revisions:

1) In Experiment 1 and also some of the following experiments, paired t-tests are used in many of the analyses. Even though these were planned comparisons, is there any correction done for multiple comparisons? In general it would have probably been better to use a GLMM and test all conditions and participants at ones, which should increase statistical power (e.g. Moscatelli et al. Journal of Vision 2012 (Modeling psychophysical data at the population-level: The generalized linear mixed model).

2) In Experiment 2 ANOVAs have been performed, but no direct comparison between the two conditions (the one with the data going up and the other with the data going down). Why not analyze all data with a between subject ANOVA and look for an interaction?

3) Figure 2E and F, were fitted with an exponential model. Please explain the choice of this model, what parameters were used and whether the parameters differ across the two conditions (which they clearly seem to do).

4) In Figure 3B you plot frequency data ranging from 0 to 1. You write you tested for normality and thus did a t-test. It seems that a more conservative, non-parametric test is more appropriate here. The data are bounded and the some of the frequencies are close to 1.

---

## [Author Response]

Summary:We all found the study to provide an interesting extension to the work on sensory gating, the experience that the sensed force of an actively produced tap is usually felt less intense than when the tactile stimulus is not the result of a volitional action. Previous studies have demonstrated that this effect is predictive, based on cancellation of the expected sensory input – and one would therefore predict it should be adaptive to changes in the contingency between action and sensation, but this has not previously been demonstrated. The experimental design to explore this question is appropriate and the results convincingly show that this gating process is subject to temporal adaptation. The finding that, after adaptation, the tactile stimulus is felt as less intense when presented at the adapted delay compared to when presented simultaneously, is especially compelling. The generalization work was also viewed as a strength of the paper, increasing the impact value.There are a few issues that should be addressed in revision. We don't foresee any of the problems being difficult to address. They mostly relate to the statistical analyses. These could, in principle, affect the conclusions that can be drawn. However, we anticipate that the data are robust enough to withstand more conservative statistical tests. This needs to be demonstrated.

We thank the editors and the reviewers for their very positive comments. We think that two points in these comments need further clarification from our side.

First, we did not study sensory gating but sensory attenuation, and we consider these to be two different phenomena.

- Sensory gating refers to the suppression of any cutaneous stimuli (both self-generated and externally generated touches/vibrations) presented on a moving limb (Chapman et al., 1987; Williams et al., 1998).

- Somatosensory attenuation refers to the suppression of exclusively self-generated stimuli (not externally generated) on a moving limb but also on a passive limb that receives a touch, such as in case of bimanual interaction (e.g., touching one hand with the other) (Bays and Wolpert, 2008).

Please note that these two tactile phenomena are different (Bays and Wolpert, 2008; Palmer et al., 2016),but unfortunately, the terms are sometimes erroneously mixed up in the literature.

Second, we did *not* show that ‘the tactile stimulus is felt as less intense when presented at the adapted delay compared to when presented simultaneously’ anywhere in our text.

- With respect to force attenuation, we showed that (a) the delayed tactile stimulus feels less intense after exposure to that delay than after exposure to a 0 ms delay, and (b) the immediate tactile stimulus feels stronger when presented without delay after exposure to a 100 ms delay than after exposure to a 0 ms delay.

- With respect to tickling, we showed that the delayed tactile stimulus is reported less frequently as the ticklish stimulus after exposure to that delay than after exposure to a 0 ms delay.

The conclusions of our manuscript are consistent with our findings, but we consider that these clarifications might be useful to future readers.

References:

Chapman, C.E., Bushnell, M.C., Miron, D., Duncan, G.H., and Lund, J.P. (1987). Sensory perception during movement in man. Exp. Brain Res. 68, 516–524.

Palmer, C.E., Davare, M., and Kilner, J.M. (2016). Physiological and Perceptual Sensory Attenuation Have Different Underlying Neurophysiological Correlates. J. Neurosci. 36, 10803–10812. Available at: http://www.jneurosci.org/lookup/doi/10.1523/JNEUROSCI.1694-16.2016.

Williams, S.R., Shenasa, J., and Chapman, C.E. (1998). Time course and magnitude of movement-related gating of tactile detection in humans. I. Importance of stimulus location. J. Neurophysiol. 79, 947–963.

Essential revisions:1) In Experiment 1 and also some of the following experiments, paired t-tests are used in many of the analyses. Even though these were planned comparisons, is there any correction done for multiple comparisons?

We did *not* perform any correction for multiple comparisons in our analyses because all our comparisons were planned based on our a priori hypotheses; these hypotheses were well grounded in earlier studies on decreased sensory attenuation with delays (Bays et al., 2005), increased ticklishness with delays (Blakemore et al., 1999), predictive motor control in the presence of delays (Witney et al., 1999) and sensorimotor adaptation to delayed visual inputs (Stetson et al., 2006).

However, we report here that all significant results from Experiment 1 and Experiment 2 remained significant when we corrected for multiple comparisons using the method for false discovery rate – FDR (Benjamini and Hochberg, 1995).

**Author response table 1. resptable1:** Experiment 1:

PSE comparisons	Original *p –* value	Corrected *p-*value
[0, 0] vs. baseline	*p* < 0.001 (0.0000676947)	*p* < 0.001 (0.0002369315)
[0, 100] vs. baseline	*p* = 0.9032264724	*p* = 0.9032264724
[0, 0] vs. [0, 100]	*p* < 0.001 (0.0000004712)	*p* < 0.001 (0.0000032987)
[0, 0] vs. [100, 0]	*p* = 0.0107374617	*p* = 0.0187905579
[100, 0] vs. baseline	*p* = 0.0701934955	*p* = 0.0818924114
[0, 100] vs. [100, 100]	*p* = 0.0026044238	*p* = 0.0060769889
[100, 100] vs. baseline	*p* = 0.0570803400	*p* = 0.0799124760

**Author response table 2. resptable2:** Experiment 2:

PSE comparisons for [100 ms, 0 ms]	Original *p* – value	Corrected *p*-value
0 vs. 50 trials	*p* = 0.1815470400	*p* = 0.1815470400
0 vs. 200 trials	*p* = 0.0282742723	*p* = 0.0424114084
0 vs. 500 trials	*p* = 0.0135470857	*p* = 0.0406412571
PSE comparisons for [100 ms, 100 ms]	Original *p* – value	Corrected *p*-value
0 vs. 50 trials	*p* = 0.0098760710	*p* = 0.0136271106
0 vs. 200 trials	*p* = 0.0136271106	*p* = 0.0136271106
0 vs. 500 trials	*p* = 0.0044865222	*p* = 0.0134595666

Reference:

Benjamini, Y., and Hochberg, Y. (1995). Controlling the False Discovery Rate: A Practical and Powerful Approach to Multiple Testing. J. R. Stat. Soc. Ser. B.

In general it would have probably been better to use a GLMM and test all conditions and participants at ones, which should increase statistical power (e.g. Moscatelli et al. Journal of Vision 2012 (Modeling psychophysical data at the population-level: The generalized linear mixed model).

We thank the reviewer(s) for this comment. We have now reanalyzed the data with a generalized linear mixed effects model (GLMM) using the mixed psychophysics toolbox MERpsychophysics (Moscatelli et al., 2012).

We first created a model with two fixed effects – a continuous effect reflecting the intensity of the comparison tap and a categorical effect reflecting the experimental condition – and one random effect, i.e., a random intercept per subject. We then contrasted this model with a second model that included the two fixed effects and two random effects, i.e., a random intercept and a random slope. When contrasting the models, it was revealed that the model with the two random effects was better; that is, the model had a lower Akaike information criterion value and a lower Bayesian information criterion value, and a chi-square test suggested that the model with two random effects was significantly better than that with only one random intercept (*p* < 0.001). Therefore, we proceeded with the model with two fixed and two random effects.

For the link function, we used the *probit* function. To estimate the Points of Subjective Equality (PSEs) of each condition, as well as the difference in PSEs between the conditions of interest, we used *bootstrapping* with B = 200 bootstrap replicates.

Author response table 3 summarizes the results. As seen, all differences are significant (confidence intervals do not include zero), except the comparison between [0 ms, 100 ms] versus baseline, which is expected not to be significant and which was also not significant in the parameter as outcome model (PAOM). In other words, the results did not change.

**Author response table 3. resptable3:** 

Comparisons	Estimate	Inferior CI	Superior CI
[0 ms, 0 ms] versus baseline	-0.124	-0.158	-0.093
[0 ms, 100 ms] versus baseline	0.020	-0.008	0.057
[0 ms, 0 ms] versus [0 ms, 100 ms]	-0.144	-0.185	-0.110
[0 ms, 0 ms] versus [100 ms, 0 ms]	-0.066	-0.104	-0.027
[0 ms, 100 ms] versus [100 ms, 100 ms]	0.087	0.051	0.125
[100 ms, 0 ms] versus baseline	-0.058	-0.093	-0.025
[100 ms, 100 ms] versus baseline	-0.067	-0.098	-0.032

We recognize the advantages of GLMMs in accounting for the within-subjects variability and in increasing the statistical power compared to a PAOM (Moscatelli et al., 2012), but we chose to keep the current analysis in our manuscript because it is the same analysis used in the previous study by Bays et al., 2005, on somatosensory attenuation and delays.

We further proceeded to improve our current modeling and statistical analysis to increase its rigorousness. Specifically,

1) We decided to apply a conservative criterion, keeping only the experimental trials in which the test tap (2 N) was within the range of [1.85, 2.15 N]. This criterion led to the rejection of few trials (2.6% in Experiment 1 and 3.6% in Experiment 2).

2) We further binned the comparison taps to their closest experimental value [1, 1.5, 1.75, 2, 2.25, 2.5, and 3 N].

3) We now report the Bayes factors for all the results in the parametric tests to provide information about the level of support for the experimental hypothesis against the null hypothesis.

By doing so, across both experiments (Experiment 1 and Experiment 2), all significant comparisons remained significant (and vice versa), except one; the comparison [100 ms, 100 ms] versus baseline became a nonsignificant trend (*p* = 0.057). However, the Bayesian analysis is inconclusive about the absence of an effect. To summarize, none of the conclusions in our manuscript changed.

Moreover, to illustrate the goodness of our model fitting, we now show all individual fits for all conditions of Experiment 1 in a new figure (Figure 2—figure supplement 1).

We have now added text in the Materials and methods to illustrate these changes (subsections “Response trials”, Data and Statistical Analysis”, “Participants, Procedure and Analysis in the Effect of Exposure Experiment”), and we have updated all statistical tests and Figure 2 to illustrate the new results (Results and Discussion, second and third paragraphs and Figure 2 legend).

Reference:

Moscatelli, A., Mezzetti, M., and Lacquaniti, F. (2012). Modeling psychophysical data at the population level: The generalized linear mixed model. J. Vis. 12, 1–1

2) In Experiment. ANOVAs have been performed, but no direct comparison between the two conditions (the one with the data going up and the other with the data going down). Why not analyze all data with a between subject ANOVA and look for an interaction?

We performed the requested mixed ANOVA with the test delay as the between-subjects factor and the exposure time as the within-subjects factor. There was no main effect of test delay (*F*(1, 56) = 0.03, *p* = 0.856), no main effect of time (*F*(3, 42) = 0.14, *p* = 0.938), and a significant interaction between these factors (*F*(3, 56) = 3.89, *p* = 0.013).

We now report this result in our manuscript (Results and Discussion, third paragraph and subsection “Participants, Procedure and Analysis I n the Effect of Exposure Experiment”).

3) Figure 2E and F, were fitted with an exponential model. Please explain the choice of this model, what parameters were used and whether the parameters differ across the two conditions (which they clearly seem to do).

We fitted the unlearning and learning curves with exponential models only for illustration purposes. We originally used the function *geom_smooth* in R, after specifying an exponential formula e-x.

The choice of the model was mainly motivated by earlier work on motor learning in space (e.g., using forcefield adaptation or visuomotor rotation) that shows that exponential models can describe the decay of the subjects’ error rates well (Brennan and Smith, 2015; Krakauer, 2009; Michel et al., 2018). In the present study, we considered that the prediction errors due to unexpected delays would also decay as the adaptation of the internal models continues.

We have now removed the exponential fits since we are not discussing them in our manuscript and we are not comparing those fits with other model fits. We have additionally corrected the x-axis to illustrate the correct distances between the ticks (0, 50, 200 and 500 trials).

References:

Brennan, A. E., and Smith, M. A. (2015). The Decay of Motor Memories Is Independent of Context Change Detection. PLoS Comput. Biol. doi:10.1371/journal.pcbi.1004278.

Krakauer, J. W. (2009). Motor learning and consolidation: The case of visuomotor rotation. Adv. Exp. Med. Biol. doi:10.1007/978-0-387-77064-2_21.

Michel, C., Bonnetain, L., Amoura, S., and White, O. (2018). Force field adaptation does not alter space representation. Sci. Rep. doi:10.1038/s41598-018-29283-z.

4) In Figure 3B you plot frequency data ranging from 0 to 1. You write you tested for normality and thus did a t-test. It seems that a more conservative, non-parametric test is more appropriate here. The data are bounded and the some of the frequencies are close to 1.

We agree with the reviewer(s), and we now report the results of a Wilcoxon signed rank test instead. The difference remained significant (*p* = 0.046) (Results and Discussion, fifth paragraph).